# Association between burnout and empathy in medical residents

**Mehrnoosh Zakerkish[1], Abdolhussein Shakurnia[2]\*, Ali Hafezi[3], Mahmood Maniati[4]**

**1** Departments of Internal Medicine, School of Medicine, Ahvaz Jundishapur University of Medical Sciences, Ahvaz, Iran, **2** Departments of Immunology, School of Medicine, Ahvaz Jundishapur University of Medical Sciences, Ahvaz, Iran, **3** General Physician, School of Medicine, Ahvaz Jundishapur University of Medical Sciences, Ahvaz, Iran, **4** Department of General Courses, School of Medicine, Ahvaz Jundishapur University of Medical Sciences, Ahvaz, Iran

\* shakurnia@yahoo.com

## Abstract

### Background

Burnout is a work-related syndrome that can affect physicians' performance. Empathy is one of the clinical competencies in whose formation many factors play a role. Empathy and burnout are important topics of research in medical sciences, and both are related to the communication between the patient and the physician. This study investigated the relationship between occupational burnout and empathy among medical residents.

### Method

This cross-sectional study was conducted on 297 medical residents in Ahvaz Jundishapur University of Medical Sciences from January to March 2021. The data collection tools were the Jefferson Scale of Empathy (JSE) and the Maslach Burnout Inventory (MBI). The reliability of the instruments was measured by calculating Cronbach's alpha. Data were analyzed by SPSS-18, using the Pearson correlation test and linear regression models.

### Results

The average age of the study population was 33.06 ±4.7 years, with more than half being females (57.6%) and married (51.9%). The residents' mean scores of empathy and overall burnout were 102.87 out of 140 and 55.90 out of 132, respectively. The burnout scores showed an inverse correlation with the overall empathy scores (r = −0.123, P = 0.035), and the score of standing in the patient's shoes (r = −0.165, P = 0.004). Linear regression test was used to detect which dimension of empathy was a better predictor for the reduction of burnout scores. Results showed that the best predictor was standing in the patient's shoes (P = 0.014).

### Conclusion

The findings showed a negative association between empathy and burnout among medical residents, which suggests the need for adjustment of the existing burnout management at

**Data Availability Statement:** The dataset/raw data presented in the study is available on request from the Research Development Unit of Ahvaz Jundishapur University of Medical Sciences (Research@ajums.ac.ir) during submission or after

publication. The data are not publicly available due to privacy ethics.

**Funding:** The author(s) received no specific funding for this work.

**Competing interests:** The authors have declared that no competing interests exist.

the institutional level. Therefore, residents should be supported by managers to reduce burnout and improve their empathic behavior.

## Introduction

Medicine is a stressful profession. It is replete with stresses that result from inconsistent and contradictory processes that affect satisfaction and occupational burnout by reducing physical and mental health and occupational involvement. Burnout is a work-related psychological problem that causes emotional problems, reduced personal success, and depersonalization [1]. Occupational burnout is one of the main problems affecting physicians' and health care staff's quality of life. High levels of burnout among physicians are a threat not only to the physicians themselves but also to their patients and the organizations they work for [2]. Burnout is a multidimensional work-related syndrome that is associated with emotional exhaustion, depersonalization, and diminution of personal accomplishment [3].

Physicians are exposed to heavy workloads, stressful work environments, and high job pressure. These working conditions lead to various types of psychological problems such as anxiety, depression, and workplace burnout in these professionals. These psychological problems have a negative effect on the doctors' quality of life and their satisfaction, which will in turn affect the quality of care they provide to patients [4]. Burnout has become a pervasive problem among medical professionals because can be highly sensitive to levels of empathy [5].

One of the factors related to burnout is empathy. Research has shown that physician empathy is an essential element in health care, which has a great impact on patient satisfaction and adherence to treatment [6]. Empathy is the emotional and cognitive ability to understand the mental states, feelings, thoughts, and motives of others and to respond appropriately to them. Empathy is one of the characteristics of interpersonal behavior that is necessary for effective communication. People with high empathy levels exhibit more prosocial behaviors while those having low empathy are characterized by more antisocial behaviors. Put in a nutshell, empathy is an essential element in the physician-patient relationship and affects patient satisfaction and appropriate clinical outcomes [7].

A number of studies have addressed the relationship between empathy and burnout [3,5,8]. Ferreira et al., for example, investigated the relationship between empathy and burnout among medical residents and specialists and found that the level of burnout increases with a decrease in empathy, and that this variable plays a significant role in predicting burnout [6]. Kharidar Atiq also drew the conclusion that there is a negative and significant relationship between empathy and job burnout among nurses [8]. In another research conducted by Yue on medical staff in a hospital in China, a negative relationship was found between empathy and job burnout [7]. According to a systematic review by Wilkinson, the majority of the reviewed studies reported a negative relationship between empathy and job burnout [3]. Of course, the results of some studies have reported a positive relationship between empathy and burnout [2,9], which needs further investigation.

Medical residents experience a great deal of stress during their training, and this stress eventually decreases their job satisfaction and increases burnout, imposing a large negative impact on their mental health and the quality of patient care [10,11]. A number of studies have shown that empathy is one of the factors that protect the mental health of medical staff against burnout [12,13]. In Iran, the relationship between empathy and burnout has already been investigated among nurses [8]. However, to the best of our knowledge and review of the

literature, no previous study has investigated the relationship between empathy and burnout among medical residents in Iran. This study was conducted in order to fulfill this gap in literature. Therefore, given the paramount importance of burnout in the mental health and empathic behavior of residents, this study aims to explore the relationship between burnout and empathy among Iranian medical residents. The findings of this study may serve as a foundation for designing mental health interventions, especially those aimed at improving resident empathy.

## Method

### Study design and participants

This was a cross-sectional study investigating the relationship between burnout and empathy in a population of medical residents in Iran. All medical residents studying at Ahvaz Jundishapur University of Medical Sciences (AJUMS) were recruited in this study (n = 472). Of all those residents, 297 completed the questionnaire (the response rate was 63.1%.). Inclusion criteria: all residents working at AJUMS. Exclusion criteria: residents who were not willing to participate. Data were collected between January 1st and March 30th, 2021. The data were collected in a hospital, following the relevant guidelines and using a paper-based process. As far as the paper-based method was concerned, a research assistant handed the participants an information sheet delineating the study's rationale and the allotted time to complete the survey. They had 10 minutes to decide whether they were willing to collaborate in the study or not. If they were willing to participate, the research assistant handed them the questionnaires. Participation in the study was voluntary, and there was no compensation for completing the questionnaire. In order to adhere to the policy of strict confidentiality, the signatures of the participants were not required, and all participants retained the right to withdraw from the research at any time without giving any reason.

### Instruments

All medical residents provided socio-demographic information and completed the validated questionnaires on burnout and empathy [9].

Burnout was measured using the Persian version of Maslach Burnout Inventory (MBI). This instrument is a scale composed of 22 items that are scored based on a 7-point Likert scale ranging from 0 (never) to 6 (everyday). The total score of the tool is obtained through the sum of the scores of all items, so the range of the total score is between 0 and 132, with higher scores indicating greater burnout. The scale measures three dimensions: emotional exhaustion, depersonalization, and lack of personal accomplishment [14]. The validity and reliability of this scale were evaluated in many studies, and it has been shown that this tool and its dimensions have sufficient validity [15–17]. The validity and reliability of this questionnaire have also been evaluated and confirmed in Iran, and its reliability coefficient has been estimated at 0.751 using Cronbach's alpha method [18]. In the present study, the reliability of the questionnaire was calculated by obtaining a Cronbach's alpha of 0.675.

Empathy was measured using the Persian version of Jefferson Scale of Physician Empathy (JSPE) The JSPE was developed and standardized by Hojat in 2002 [19]. This instrument is a scale composed of 20 items that are scored based on a 7-point Likert scale ranging from 1 (strongly disagree) to 7 (strongly agree). The total score of the tool is obtained through the sum of the scores of all items, so the range of the total score is between 20 and 140, with higher scores representing greater empathy. This questionnaire has three dimensions: Perspective Taking, Compassionate Care, and Walking in Patient's Shoes. The validity and reliability of this questionnaire have been confirmed in numerous studies [19–21]. In Iran, Shariat et al.

reported a Cronbach's alpha coefficient of 0.88 for the Persian version of this questionnaire [21]. In the present study, the reliability of JSPE was calculated obtaining a Cronbach's alpha of 0.766.

### Ethical and confidentiality considerations

The study was approved by the Ethics Committee of the Ahvaz Jundishapur University of Medical Sciences (Ref. ID: IR.AJUMS.REC.1399.420). After obtaining the necessary permits for sampling, the researchers visited the teaching hospitals, and after explaining the objectives of the study to the residents and obtaining their consent to participate in the study, the mentioned questionnaires were given to them. Participation in the study was completely optional, the questionnaires were anonymous, and the residents were assured of the confidentiality of their information.

### Statistical analysis

The normality of data distribution was checked using the Kolmogorov-Smirnov test which indicated normal distribution of data. The reliability of the MBI and JSPE was measured by calculating Cronbach's alpha. Data were analyzed using the Statistical Package for the Social Sciences (SPSS) (Version 18.0). We described the numerical data by mean and standard deviation (SD), and nominal data was represented as raw counts (n) and percentages (%). The burnout and empathy scores of the residents were calculated. In order to investigate the relationship between burnout and the demographic variables, Pearson's correlation coefficient and linear regression model were used. P values < 0.05 were considered statistically significant.

## Results

### Demographic characteristics

A total of 297 residents with a mean age of 33.06 ±4.7 years participated in the present study. More than half of the respondents were female (57.6%) and married (51.9%). The mean burnout score of the residents was 55.90±13.25 while their mean empathy score was 102.13±25. The mean scores of burnout and empathy along with their dimensions are displayed in Table 1.

Table 2 shows the correlation coefficients between the residents' mean burnout and empathy scores and between dimensions of burnout and empathy. The results show that there is a significant negative relationship between burnout and empathy (r = -0.123; p = 0.035). The correlation coefficients between burnout and empathy dimensions were also negative.

**Table 1. Mean scores of burnout and empathy along with their dimensions among medical residents.**

| Variables | | Mean | SD | minimum | maximum |
|---|---|---|---|---|---|
| **Empathy** | | **102.78** | **12.73** | **67** | **134** |
| dimensions of Empathy | Perspective taking | 54.82 | 7.27 | 33 | 70 |
| | Compassionate care | 38.32 | 6.46 | 18 | 54 |
| | Walking in the patient's shoes | 9.64 | 2.58 | 2 | 14 |
| **Burnout** | | **55.90** | **13.25** | **19** | **113** |
| dimensions of Burnout | Emotional exhaustion (EE) | 20.38 | 11.52 | 0 | 51 |
| | Depersonalization (DP) | 7.21 | 5.24 | 0 | 24 |
| | Personal accomplishment (PA) | 28.36 | 7.10 | 4 | 47 |

**Table 2. Correlation matrix between burnout and empathy and between their dimensions.**

| Variable | | | Empathy | Three dimensions of Empathy | | |
|---|---|---|---|---|---|---|
| | | | | Perspective taking | Compassionate care | Walking in the patient's shoes |
| Burnout | | r | 0.123- | - 0.110 | - 0.052 | - 0.165 |
| | | p | 0.035 | 0.0580 | 0.37 | 0.004 |
| Three dimensions of Burnout | Emotional Exhaustion | r | -.166** | -.121* | -.139* | -.128* |
| | | p | 0.004 | 0.0370 | 0.016 | 0.027 |
| | Depersonalization | r | -.378** | -.309** | -.242** | -.385** |
| | | p | 0.0001 | 0.00010 | 0.0001 | 0.0001 |
| | Lack of Personal Accomplishment | r | .333** | .230** | .316** | .279** |
| | | p | 0.0001 | 0.00010 | 0.0001 | 0.0001 |

However, this relationship was significant only in the sub-scale of "Walking in Patient's Shoes" (r = -0.165; p = 0.004). The correlation coefficients of EE and DP dimensions of burnout with empathy and its dimensions were also negative. However, the relationship of PA dimensions of burnout with empathy and its dimensions was positive (Table 2).

Linear regression was used to investigate the relationship between burnout and empathy and between their dimensions and to see to what extent and in what direction empathy and its dimensions predict burnout. The results of the linear regression equation are given in Table 3. As the findings of this table show, empathy is a negative and significant predictor of burnout.

According to Table 3, the results of linear regression analysis indicate that the explanatory variables (empathy dimensions) can significantly predict and explain changes in the dependent variable (residents' burnout). In other words, the model is meaningful, and among the three dimensions of empathy (i.e., Perspective Taking, Compassionate Care, and Walking in Patient's Shoes) the most determining component is Walking in Patient's Shoes. This sub-scale is a negative and significant predictor of job burnout and explains a total of 2.3% of the variance of burnout.

The results also showed that among the three burnout dimensions, emotional exhaustion has the least relationship with empathy dimensions. Also, none of the dimensions of empathy

**Table 3. Predicting burnout through empathy dimensions in residents.**

| Dependent Variable | Predictive Variable | B | Beta | t | P value | $R^2$ | F |
|---|---|---|---|---|---|---|---|
| Burnout | Perspective taking | -0.147 | 0.081- | -1.252 | 0.212 | 0.032 | 3.264 |
| | Compassionate care | 0.061 | -0.030 | 0.495 | 0.647 | | |
| | Walking in the patient's shoes | -0.778 | -0.152 | -2.471 | 0.014 | | |
| Emotional Exhaustion | Perspective taking | -0.096 | -0.061 | -0.942 | 0.347 | 0.030 | 3.048 |
| | Compassionate care | -0.156 | -0.088 | -1.347 | 0.179 | | |
| | Standing in the patient's shoes | -0.373 | -0.084 | -1.361 | 0.175 | | |
| Depersonalization | Perspective taking | -0.141 | -0.196 | -3.337 | 0.001 | 0.195 | 23.656 |
| | Compassionate care | -0.051 | -0.063 | -1.055 | 0.292 | | |
| | Walking in the patient's shoes | -0.630 | -0.310 | -5.543 | 0.0001 | | |
| Lack of Personal Accomplishment | Perspective taking | 0.097 | 0.100 | 1.624 | 0.105 | 0.120 | 13.276 |
| | Compassionate care | 0.268 | 0.244 | 3.929 | 0.0001 | | |
| | Walking in the patient's shoes | 0.268 | 0.098 | 1.666 | 0.097 | | |
| Burnout | Empathy | -0.128 | -0.123 | -2.123 | 0.035 | 0.015 | 4.506 |

could act as a predictive factor to predict emotional exhaustion in residents. However, the two dimensions of DP and PA could be predicted by the empathy dimensions.

The regression model showed that the empathy dimensions could predict the DP sub-scale of burnout and explain 19.5% of the burnout variance. However, as far as the PA subscale is concerned, empathy dimensions could predict burnout and positively explain 12% of the variance of burnout (Table 3).

## Discussion

This study investigated the relationship between empathy and burnout in medical residents. According to our results, the mean empathy score of the residents was 102.78 ± 12.73, with the lowest and highest possible scores of empathy being 20 and 140, respectively. These results indicate that the studied residents' level of empathy is above average, which is in line with the findings of Shariat [21] and Aziz [22], who reported an average level of empathy among the residents in their study. However, in other studies, the level of residents' empathy level has been reported to be lower or higher than average [12,23,24]. These discrepancies can be attributed to differences in education and health care systems, and to cultural differences that shape patients' expectations of an "ideal physician."

In this study, the mean burnout score of the studied residents was 55.90±13.25, with the minimum and maximum scores of burnout ranging between 0 and 132. These results indicate that the level of burnout among the residents is lower than average. The literature suggests that many physicians are at risk of burnout or are currently suffering from it. In line with the results of the present study, a systematic review reported the level of burnout in the medical staff to be at an average level [25]. Passos in Brazil reported a high level of burnout among medical residents [26]. In a similar study in Nigeria, a severe degree of burnout in residents was reported [27]. Medical residents face stressful situations during their training, especially when they are in clinical settings. These include acute diseases, accidents, death, and night shifts, which require planning and changes in the curriculum to improve their quality of life and reduce burnout.

In the present study, a negative correlation was found between empathy and burnout scores. Specifically, an increase in the empathy score led to a decrease in the burnout score. These results are similar to previous reports in other populations [6,12,28–30], which means that when physicians are at a high level of burnout, their relationship with their patients seems to worsen. The empathy between the patient and the physicians promotes mutual trust and thus the effectiveness of the treatment. In a study investigating the role of empathy in burnout, Wagman et al. reported that higher levels of empathy are associated with lower levels of burnout [29]. They maintained that given the fact that low empathy can make people more vulnerable to burnout and that strengthening empathy skills can reduce or prevent burnout, it is better to include empathy in medical education programs.

The findings of this study are consistent with those of Yu [7], Khalid [28], and Lee [30] who reported a negative and significant relationship between empathy and burnout. In a systematic review, Wickenson et al. reported a negative relationship between empathy scores and burnout in most studies conducted on healthcare staff [3]. Wang et al. also reported that empathy has a positive relationship with job satisfaction but a negative association with burnout [4]. To explain this finding, it can be argued that empathy is essential for successful interpersonal functioning, which provides the basis for improving proper relationships with others. If interpersonal relationships and empathy among employees are strengthened, their resilience against difficult situations will automatically increase, and their burnout will experience a downward trend. Examining the relationship between burnout and empathy, Zenasni et al.

noted that empathy protects doctors from burnout [31]. According to their study, empathy requires awareness of negative emotions and obliges the doctor to understand patients' mental and psychological conditions through positive interactions. These interactions are resources to deal with stress and burnout at the workplace. Therefore, helping physicians practice more empathy can help protect them from burnout [32].

Despite the compelling evidence indicating a negative relationship between empathy and burnout, there are also studies reporting a positive correlation between these two variables [3,9]. This hypothesis is somewhat consistent with Maslach and Jackson's theory according to which empathetic employees are more likely to experience burnout [3]. Considering burnout as a cross-cultural construct, the available evidence emphasizes that the language in which the questionnaire is written may affect the way participants respond to its questions, which will lead to different results. Certainly, further research will shed more light on this topic.

In a systematic review of studies conducted on the relationship between empathy and burnout, Delgado et al. reported that burnout is somewhat associated with depersonalization. In fact, when human aspects of social interactions are reduced, this is likely to lead to a significant reduction in empathy. Hence, burnout is fatal for both the doctor and the patient. Exhausted doctors are less likely to have the ability to stand in the patient's shoes and listen actively. Instead, they prefer to protect themselves by distancing themselves from patients and depersonalizing them. Delgado et al. maintain that the higher the burnout level of physicians, the lower their clinical empathy [13]. Investigating and understanding the complex links between empathy and burnout can help healthcare professionals as well as educational institutions to reduce the risk of burnout.

Our analysis of the relationship between burnout and empathy subscales showed that out of the three empathy subscales, only "Walking in Patient's Shoes" had a significant relationship with burnout, and this subscale can act as a predictive factor and predict burnout in residents. Based on this, residents who had lower empathy levels were associated with higher levels of burnout. In other words, the ability to "walk in the patient's shoes" provides a protective effect for the residents against burnout. Rodríguez-Nogueira et al. suggest that burnout plays a challenging role in clinical empathy and that burnout can affect empathy levels by creating job dissatisfaction [33].

The findings of this study show the discrete importance of different empathy subscales because their impact on burnout subscales is different. Among the three burnout subscales, emotional exhaustion showed the lowest correlation with empathy and its subscales. Examining the relationship between burnout subscales and empathy subscales showed that there is a significant negative relationship between empathy subscales and emotional exhaustion. However, none of the subscales of empathy has the power to predict emotional exhaustion in the residents. There was a negative and significant relationship between the subscale of depersonalization and empathy and its subscales. Also, the two subscales of "Perspective Taking" and "Walking in Patient's Shoes" could predict the depersonalization subscale in the residents. The results of our regression analysis revealed that these two variables together could predict burnout in a negative direction with a determination coefficient of 19.5% of the variance. Regarding the relationship of the subscale of "lack of personal accomplishment" with empathy and its subscales, it was observed that there is a positive relationship between these variables and that the subscales of "Perspective Taking" and "Compassionate care" could predict the subscale of "Lack of personal accomplishment" and explain a total of 12% of the variance. In a systematic review, the relationship of different components of empathy with each component of burnout in doctors was investigated [13]. The results showed that the subscales of emotional exhaustion and depersonalization had a negative relationship with the subscales of "Perspective-taking", "Compassionate care", and "Walking in the patient's shoes. There was also a positive

relationship between the subscale of "Lack of personal accomplishment" and the subscales of empathy, and high levels of "Lack of personal accomplishment" were associated with high levels of "Perspective-taking", "Compassionate care", and "Walking in the patient's shoes", which is consistent with the findings of our study.

Residents should be aware that dealing with patients' pain and suffering on a regular basis puts their emotional health at risk, and that experiencing strong emotions and feelings may affect their personal life. Therefore, having an understanding of burnout and dealing with it as a reality is essential to a resident's career survival. Residents must learn how to protect themselves in this way. A previous study found that residents who had lower empathy scores also had a lower quality of life [34]. Also, in another study, a significant relationship was found between the scores of quality of life and burnout [35]. In a content analysis study, Ahmadian et al. reported that burnout plays a challenging role in clinical empathy [36]. They showed that lack of coping strategies may create a high level of stress and burnout and that extracurricular educational activities, which are considered as coping mechanisms, can have a good effect on mental and physical health and reduce stress and burnout. In order to reduce job burnout and increase the quality of life, planning for extracurricular activities should be done in order to increase coping mechanisms against stress and burnout in students. Managers of residency curricula should also be sensitive to the need for the inclusion of training programs for teaching empathic strategies and emotional management for the professional development of medical residents. This will improve the mental health of residents and enhance the quality of medical services by reducing the rate of burnout.

This research had a number of limitations. The main limitation was that this study was conducted only in one university, which limits the generalizability of the results. Another limitation is that data collection was done through convenience sampling, a method that can lead to biases in data collection and sampling. Therefore, caution should be exercised when interpreting the results, paying attention to the fact that the participants in the study were not selected randomly and were not from several universities. Additionally, we did not study other factors that may influence levels of empathy or burnout. These include personality traits, prior psychiatric disorders, or a history of using medication. Further studies should be conducted to determine other factors influencing empathic capacity during the physician's career.

## Conclusion

The findings of this study showed a negative association between empathy and burnout among medical residents, which suggests the need for adjustment of the existing burnout management practices at the institutional level. Of all dimensions of empathy, "Walking in the patient's shoes" had the most important association with reduced burnout scores and was also a predicting variable for burnout. Medical educators must take these findings into account when planning innovative strategies to promote empathy among medical residents. Therefore, residents should be supported by managers to reduce burnout and improve empathic behavior.

## Acknowledgments

The authors wish to thank the medical residents who participated in the project and completed the questionnaire.

## Author Contributions

**Conceptualization:** Mehrnoosh Zakerkish, Abdolhussein Shakurnia, Ali Hafezi.

**Data curation:** Ali Hafezi.

**Formal analysis:** Mahmood Maniati.

**Investigation:** Ali Hafezi.

**Methodology:** Abdolhussein Shakurnia, Mahmood Maniati.

**Project administration:** Abdolhussein Shakurnia.

**Software:** Ali Hafezi.

**Supervision:** Mehrnoosh Zakerkish, Abdolhussein Shakurnia.

**Writing – original draft:** Abdolhussein Shakurnia, Mahmood Maniati.

**Writing – review & editing:** Mehrnoosh Zakerkish, Abdolhussein Shakurnia, Mahmood Maniati.

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
