## [Decision Letter · Decision Letter 0]

27 Sep 2023

PONE-D-23-22729Association between Burnout and Empathy in medical residentsPLOS ONE

Dear Dr. shakurnia,

Thank you for submitting your manuscript to PLOS ONE. After careful consideration, we feel that it has merit but does not fully meet PLOS ONE’s publication criteria as it currently stands. Therefore, we invite you to submit a revised version of the manuscript that addresses the points raised during the review process.

ACADEMIC EDITOR:- Please provide strong justification of the importance of your research and difference between it and other similar studies specially in your country.- The study design : it's known that the  observational correlational studies include; cross sectional studies, case control studies, cohort studies. As your study is suitable to be crooss sectional study design, so I prefere to be more specific in mentioning the study design as cross section study.- The method part need to be more oraganized as; study design, timing and setting, participant, sampling, tools of data collection, ethical consideration and data mangment and analysis.- Please mention the responce rate of the particpants in the method section.- Did you conduct a pilot study?

We look forward to receiving your revised manuscript.

Kind regards,

Omnia Samir El Seifi, Professor

Academic Editor

PLOS ONE

Journal Requirements:

Reviewers' comments:

Reviewer's Responses to Questions

**Comments to the Author**

1. Is the manuscript technically sound, and do the data support the conclusions?

Reviewer #1: Partly

Reviewer #2: Partly

2. Has the statistical analysis been performed appropriately and rigorously? 

Reviewer #1: Yes

Reviewer #2: Yes

3. Have the authors made all data underlying the findings in their manuscript fully available?

Reviewer #1: Yes

Reviewer #2: Yes

4. Is the manuscript presented in an intelligible fashion and written in standard English?

Reviewer #1: No

Reviewer #2: No

5. Review Comments to the Author

Reviewer #1: The research question has been widely addressed by recent published literature and fails to add to the existing knowledge base. Examples include:

Pitanupong J, Sathaporn K, Ittasakul P, Karawekpanyawong N. Relationship of mental health and burnout with empathy among medical students in Thailand: A multicenter cross-sectional study. Plos One. 2023 Jan 5;18(1):e0279564.

Picard J, Catu-Pinault A, Boujut E, Botella M, Jaury P, Zenasni F. Burnout, empathy and their relationships: a qualitative study with residents in General Medicine. Psychology, health & medicine. 2016 Apr 2;21(3):354-61.

Thirioux B, Birault F, Jaafari N. Empathy is a protective factor of burnout in physicians: new neuro-phenomenological hypotheses regarding empathy and sympathy in care relationship. Frontiers in psychology. 2016 May 26;7:763.

Reviewer #2: Dear Author, Though it is an interesting research, there is always room for improvement. You may kindly consider the following points:

1) Thoroughly revise your manuscript and read your Abstract at two places in your article and make sure that both texts are the same and present in the main article as well. Try to improve your wording.

2)Give further strength to your rationale/purpose of study.

3)In methods, too much detail is given under MEASURES section. Try to add your own performed METHODS rather than giving a literature review. Consider renaming your study design. Mention type of regression used. Exclusion criteria has some not-included variables which may be deleted. If it is a census then there is no need to use the word SAPMLE or sampling technique.

4) Reconsider your tables and make it more comprehensible and in standard formats especially table 1 and table 4.Table 3 can be adjusted within table 4. Kindly get guidance from other published researches how to report regression analysis.

5) Rewrite a short but strong conclusion keeping in view your study objective and results.

6. PLOS authors have the option to publish the peer review history of their article (what does this mean?). If published, this will include your full peer review and any attached files.

Reviewer #1: No

Reviewer #2: No

---

## [Author Response · Author response to Decision Letter 0]

13 Jan 2024

ACADEMIC EDITOR:

- Please provide strong justification of the importance of your research and difference between it and other similar studies specially in your country.

According to the reviewer's comment, corrections were made in the final paragraph of the introduction section.

- The study design: it's known that the observational correlational studies include; cross sectional studies, case control studies, cohort studies. As your study is suitable to be crooss sectional study design, so I prefere to be more specific in mentioning the study design as cross section study.

In the manuscript, the study design was to cross-sectional.

- The method part need to be more oraganized as; study design, timing and setting, participant, sampling, tools of data collection, ethical consideration and data mangment and analysis.

The method section was organized according to the reviewer's comment.

- Please mention the response rate of the participants in the method section.

According to the reviewer recommendation, the response rate was transferred to the method section.

- Did you conduct a pilot study?

No

Journal Requirements:

Our manuscript meets PLOS ONE's style requirements, including those for file naming. The PLOS ONE style templates.

The data were collected in a hospital following relevant guidelines and using a paper-based process. As far as the paper-based method was concerned, a research assistant handed the participants an information sheet delineating the study's rationale and the allotted time to complete the survey. They had 10 minutes to decide whether they were willing to collaborate in the study or not. If they were, the research assistant would hand them the questionnaires. Participation in the study was voluntary, and there was no compensation for completing the questionnaire. Adhering to the policy of strict confidentiality, the signatures of the participants were not required, and all the participants retained the right to withdraw from the research at any time without giving any reason.

All data are available by corresponding author via Email (shakurnia@yahoo.com) upon reasonable request.

Reviewers' comments:

Reviewer's Responses to Questions

Comments to the Author

1. Is the manuscript technically sound, and do the data support the conclusions?

Reviewer #1: Partly

Reviewer #2: Partly

2. Has the statistical analysis been performed appropriately and rigorously? 

Reviewer #1: Yes

Reviewer #2: Yes

3. Have the authors made all data underlying the findings in their manuscript fully available?

Reviewer #1: Yes

Reviewer #2: Yes

4. Is the manuscript presented in an intelligible fashion and written in standard English?

Reviewer #1: No

Reviewer #2: No

5. Review Comments to the Author

Reviewer #1: The research question has been widely addressed by recent published literature and fails to add to the existing knowledge base. Examples include:

Pitanupong J, Sathaporn K, Ittasakul P, Karawekpanyawong N. Relationship of mental health and burnout with empathy among medical students in Thailand: A multicenter cross-sectional study. Plos One. 2023 Jan 5;18(1):e0279564.

Picard J, Catu-Pinault A, Boujut E, Botella M, Jaury P, Zenasni F. Burnout, empathy and their relationships: a qualitative study with residents in General Medicine. Psychology, health & medicine. 2016 Apr 2;21(3):354-61.

Thirioux B, Birault F, Jaafari N. Empathy is a protective factor of burnout in physicians: new neuro-phenomenological hypotheses regarding empathy and sympathy in care relationship. Frontiers in psychology. 2016 May 26;7:763.

Amendments were made in the introduction section in order to satisfy the request of the reviewer's comment.

Reviewer #2: Dear Author, Though it is an interesting research, there is always room for improvement. You may kindly consider the following points:

1) Thoroughly revise your manuscript and read your Abstract at two places in your article and make sure that both texts are the same and present in the main article as well. Try to improve your wording.

The abstract was reviewed and revised.

2)Give further strength to your rationale/purpose of study.

The rationale and purpose of the study were reviewed and completed.

3)In methods, too much detail is given under MEASURES section. Try to add your own performed METHODS rather than giving a literature review. Consider renaming your study design. Mention type of regression used. Exclusion criteria has some not-included variables which may be deleted. If it is a census then there is no need to use the word SAPMLE or sampling technique.

In the method section, corrections were made according to the reviewer's comment, and the manuscript was corrected based on the reviewer’s comments.

4) Reconsider your tables and make it more comprehensible and in standard formats especially table 1 and table 4. Table 3 can be adjusted within table 4. Kindly get guidance from other published researches how to report regression analysis.

The tables were revised and Table 3 was merged into Table 4.

5) Rewrite a short but strong conclusion keeping in view your study objective and results.

The conclusion was reviewed and revised in the abstract and in the final paragraph of the manuscript.

---

## [Decision Letter · Decision Letter 1]

7 Feb 2024

PONE-D-23-22729R1Association between Burnout and Empathy in medical residentsPLOS ONE

Dear Dr. shakurnia,

Thank you for submitting your manuscript to PLOS ONE. After careful consideration, we feel that it has merit but does not fully meet PLOS ONE’s publication criteria as it currently stands. Therefore, we invite you to submit a revised version of the manuscript that addresses the points raised during the review process.

ACADEMIC EDITOR: />==============================

We look forward to receiving your revised manuscript.

Kind regards,

Omnia Samir El Seifi, M.D.

Academic Editor

PLOS ONE

Journal Requirements:

Additional Editor Comments:

Reviewers' comments:

Reviewer's Responses to Questions

Comments to the Author

1. If the authors have adequately addressed your comments raised in a previous round of review and you feel that this manuscript is now acceptable for publication, you may indicate that here to bypass the “Comments to the Author” section, enter your conflict of interest statement in the “Confidential to Editor” section, and submit your "Accept" recommendation.

Reviewer #1: All comments have been addressed

Reviewer #2: All comments have been addressed

2. Is the manuscript technically sound, and do the data support the conclusions?

Reviewer #1: Yes

Reviewer #2: Yes

3. Has the statistical analysis been performed appropriately and rigorously? 

Reviewer #1: Yes

Reviewer #2: Yes

4. Have the authors made all data underlying the findings in their manuscript fully available?

Reviewer #1: Yes

Reviewer #2: (No Response)

5. Is the manuscript presented in an intelligible fashion and written in standard English?

Reviewer #1: Yes

Reviewer #2: No

6. Review Comments to the Author

Reviewer #1: Thank you for addressing the points highlighted in the previous review. The manuscript reads well and easy to interpret and adds value to existing knowledge in this area of research.

Reviewer #2: Thanks for incorporating most of the recommended changes. You may kindly note the following MINOR changes please:

1-Download the attached pdf and incorporate COMMENTS given for highlighted words/sentence

2-Kindly correct grammatical mistakes through proof reading by a language specialist.

7. PLOS authors have the option to publish the peer review history of their article (what does this mean?). If published, this will include your full peer review and any attached files.

Do you want your identity to be public for this peer review? For information about this choice, including consent withdrawal, please see our Privacy Policy.

Reviewer #1: No

Reviewer #2: No

---

## [Author Response · Author response to Decision Letter 1]

19 Feb 2024

Dear Omnia Samir El Seifi, M.D.

Academic Editor PLOS ONE, 

PONE-D-23-22729. Association between Burnout and Empathy in Medical Residents

It is with great pleasure that we resubmit to you a revised version of manuscript entitled "Association between Burnout and Empathy in Medical Residents". We would like to thank you for giving us the opportunity to revise and resubmit this manuscript. We appreciate the time and care you and the reviewers took in providing us with the valuable comments, and we have incorporated the suggested changes into the manuscript to the best of our ability. 

The edited file of the manuscript is sent to you in two forms, a clean and Revised Manuscript with Track Changes.

The reference list was checked. The correctness of the reference is confirmed.

With regards,

Corresponding author

Abdolhussein Shakurnia

---

## [Decision Letter · Decision Letter 2]

19 Mar 2024

Association between Burnout and Empathy in medical residents

PONE-D-23-22729R2

Dear Dr. shakurnia,

We’re pleased to inform you that your manuscript has been judged scientifically suitable for publication and will be formally accepted for publication once it meets all outstanding technical requirements.

An invoice for payment will follow shortly after the formal acceptance. To ensure an efficient process, please log into Editorial Manager at Editorial Manager® , click the 'Update My Information' link at the top of the page, and double check that your user information is up-to-date. If you have any billing related questions, please contact our Author Billing department directly at authorbilling@plos.org.

Kind regards,

Omnia Samir El Seifi, M.D.

Academic Editor

PLOS ONE

Additional Editor Comments (optional):

Reviewers' comments:

Reviewer's Responses to Questions

**Comments to the Author**

1. If the authors have adequately addressed your comments raised in a previous round of review and you feel that this manuscript is now acceptable for publication, you may indicate that here to bypass the “Comments to the Author” section, enter your conflict of interest statement in the “Confidential to Editor” section, and submit your "Accept" recommendation.

Reviewer #2: All comments have been addressed

2. Is the manuscript technically sound, and do the data support the conclusions?

Reviewer #2: Yes

3. Has the statistical analysis been performed appropriately and rigorously? 

Reviewer #2: Yes

4. Have the authors made all data underlying the findings in their manuscript fully available?

Reviewer #2: Yes

5. Is the manuscript presented in an intelligible fashion and written in standard English?

Reviewer #2: Yes

6. Review Comments to the Author

Reviewer #2: dear Author, Thanks for incorporating suggested changes. It would be better if you assign line numbers to your manuscript in future. It makes review very easy and time saving.

7. PLOS authors have the option to publish the peer review history of their article (what does this mean?). If published, this will include your full peer review and any attached files.

Reviewer #2: No

---

## [Editor Report · Acceptance letter]

1 Apr 2024

PONE-D-23-22729R2 

PLOS ONE

Dear Dr. Shakurnia, 

I'm pleased to inform you that your manuscript has been deemed suitable for publication in PLOS ONE. Congratulations! Your manuscript is now being handed over to our production team.

Kind regards, 

on behalf of

Professor Omnia Samir El Seifi 

Academic Editor

PLOS ONE